# CCIL: Context-conditioned imitation learning for urban driving

## Abstract

Imitation learning is a promising solution to the challenging autonomous urban driving task as experienced human drivers can effortlessly tackle highly complex driving scenarios. Behavior cloning is the most widely applied imitation learning approach in autonomous driving due to its exemption from potentially risky online interactions, but it suffers from the covariate shift issue. To mitigate this problem, we propose a context-conditioned imitation learning approach that learns a policy to map the context state into the ego vehicle's state instead of the typical formulation from both ego and context state to the ego action. Besides, to make full use of the spatial and temporal relations in the context to infer the ego future states, we design a novel policy network based on the Transformer, whose attention mechanism has demonstrated excellent performance in capturing relations. Finally, during evaluation, a linear quadratic controller is employed to produce smooth planning based on the predicted states from the policy network. Experiments on the real-world large-scale Lyft and nuPlan datasets demonstrate that our method can surpass the state-of-the-art method significantly.

## 1 Introduction

Planning a safe, comfortable, and efficient trajectory in a complex urban environment for a self-driving vehicle (SDV) is an important and challenging task in autonomous driving (Yurtsever et al., 2020). Unlike highway driving (Henaff et al., 2019), urban driving requires handling more varied road geometry such as roundabouts and intersections while interacting with traffic lights, pedestrians, and other vehicles. Classic manually-designed rule-based approaches (Fan et al., 2018) have achieved some success in industry but demand tedious human engineering to struggle with diverse real-world cases. Meanwhile, the rapid development of deep learning techniques motivates researchers (Bojarski et al., 2016; Pan et al., 2020) to employ a deep neural network to model the complicated driving policy. To learn such a policy, imitation learning (IL) from human drivers' demonstrations is a promising solution as experienced drivers can tackle even extremely challenging situations, and their driving data can be collected at scale.

The simplest IL algorithm is the behavior cloning (BC) method, which has wide applications in autonomous driving (Pomerleau, 1988; Bojarski et al., 2016; Codevilla et al., 2018b) due to its exemption from potentially dangerous online interactions. It learns a policy in a supervised fashion by minimizing the difference between the learner's action and the expert's action in the expert state distribution. However, the BC method suffers from the *covariate shift* issue (Ross et al., 2011), *i.e.* the state induced by the learner's policy cumulatively deviates from the expert's distribution.

To overcome the *covariate shift* obstacle, existing methods such as DAgger (Ross et al., 2011) and DART (Laskey et al., 2017) query supervisor corrections at the learner's states or perturbed expert's states. Since human supervision is hard to collect, recent works like GAIL (Ho & Ermon, 2016) seek to provide feedback from a neural network-based discriminator to recover from out-of-distribution states generated by the learner's policy. However, these data augmentation methods need either expert supervision or rolling out their policy in the real world or a realistic simulator, which are impractical in autonomous driving. Instead, some researchers attempt to constrain the learned policy formulation to ensure its robustness to the policy error by incorporating control theoretic prior knowledge. For example, Palan et al. (2020); Havens & Hu (2021) pose Kalman or linear matrix inequality constraints on the learned linear policy to guarantee its closed-loop stability in a linear

time-invariant (LTI) system. Yin et al. (2021) relaxes the linear policy formulation into a simple feed-forward neural network representation and East (2022) extends the method in Havens & Hu (2021) to polynomial policy and dynamical system. However, the urban driving task is too complex to be handled by these naive policy formulations.

To learn a stable and general urban driving policy by imitating only the offline human demonstrations, we propose a context-conditioned imitation learning (CCIL) method, where a policy network is learned to predict the SDV's future states using only its observed context, different from classic policy taking both ego state and context as input to generate its action (Codevilla et al., 2019; Bansal et al., 2019). The motivation is that the ego state can be easily influenced by the policy's error, thus leading to the catastrophic distribution shift. On the contrary, the static context elements such as lanes or crosswalks will not be influenced by the SDV and the dynamic context element like human drivers will try to recover from its perturbation. Thus, based on the stability assumption of the traffic system, we can prove theoretically that our policy formulation can achieve closed-loop stability, thus addressing the distribution shift issue. In practice, as it becomes challenging to accurately plan a trajectory for a SDV without its historical trajectory, we construct our policy network based on Transformer (Vaswani et al., 2017) to exploit spatial and temporal relation information in highly interactive and constrained urban driving scenarios. Furthermore, during evaluation, we employ a linear-quadratic regulator (LQR) (Åström & Murray, 2021) to yield a smooth action based on the SDV's current and predicted states.

The main contributions of this paper can be summarized as follows:

1. To address the *covariate shift* issue in offline imitation learning, we propose a novel context-conditioned imitation learning method, where a policy is learned to output the ego state using only its context as input. A robustness assurance is provided for our policy formulation based on an assumption of the context's stability.

2. To apply our method to urban driving, we remove the explicit ego state information input and propose a new ego-perturbed goal-oriented coordinate system to reduce the implicit ego information in the coordinate system. Besides, we design a Transformer-based planning network to make full use of the spatial and temporal information in the context.

3. To verify the effectiveness of our approach, we benchmark the real-world large-scale urban driving Lyft (Houston et al., 2020) and nuPlan (Caesar et al., 2021) datasets with state-of-the-art performance. The video and code can be found at `https://sites.google.com/view/contextconditionedil`.

## 2 RELATED WORK

### 2.1 IMITATION LEARNING FOR AUTONOMOUS DRIVING

The objective of applying IL in autonomous driving is to learn driving behavior mimicking human drivers. The most straightforward solution is BC which minimizes the difference between the learner's and the expert's action on the expert state without demanding extra manually labeled data and online interaction. Early BC applications in autonomous driving such as ALVINN (Pomerleau, 1988) and PilotNet (Bojarski et al., 2016) learn an end-to-end policy that directly maps sensor inputs to vehicle control commands using a large amount of human driving experience. Recently, ChauffeurNet (Bansal et al., 2019) provides intermediate planning using perception results to improve generalization and transparency. However, the BC approach typically leads to the *covariate shift* between the training distribution and deploying distribution, as minor errors in the policy can lead to deviating from the expert state and then larger errors.

IL methods to address the distribution shift challenge can be categorized into online methods and offline model-free and model-based methods. Online methods try to directly match the expert state-action distribution instead of matching the expert state conditioned action distribution like BC. For instance, the method in Zhang & Cho (2017) based on DAgger (Ross et al., 2011) queries supervisor actions at the state the learner visits and then adds the new data into the dataset, thus adjusting the expert state distribution to match the learner's state distribution. To get rid of the requirement of interactive expert, several methods like Wang et al. (2021) based on adversarial imitation learning (Ho & Ermon, 2016) utilize a discriminator to measure the difference between learner and expert's state-

action distribution and then compute the reward in reinforcement learning. By raising the policy's accumulated reward, the policy's state distribution will get close to the expert distribution. However, the reinforcement loop needs online interaction with the environment to generate the policy's state distribution, which is hard to deploy in safe-critical tasks like autonomous driving. On the contrary, our method matches ego state distribution conditioned on context, which can be learned by supervised learning without interaction with the environment or access to expert supervision.

The most popular model-free IL methods are based on DART (Laskey et al., 2017) which avoids the compounding error by providing synthetic examples of how to recover from the deviated state. In Codevilla et al. (2018b), they inject temporally correlated noise into the trajectory to simulate gradual drift away from the desired trajectory. Alternatively, ChauffeurNet (Bansal et al., 2019) adds a uniform perturbation to the current SDV state and fits a new smooth trajectory that brings the SDV back to the original target location. However, these rule-based trajectory augmentation methods are hard to cover the real motion distribution induced by the learner's policy and the policy is very likely to learn a propensity for perturbed driving. Our method also applies perturbation to the SDV's current position but its role is to blur the ego position information instead of data augmentation. Therefore, our method does not have the trajectory smoothing process during training. Our method is also model-free but we seek to endow the policy with robustness properties by constraining its formulation without resorting to recovery examples.

Model-based IL methods address the distribution shift by minimizing the difference between a trajectory rolling out in a differentiable learned or data-driven model with the expert trajectory. PPUU (Amos et al., 2018) first learn a dynamics model based on variational autoencoder (Kingma & Welling, 2013) from data and then train the policy network to output actions that lead to a similar trajectory as the expert trajectory. Since the dynamics model is differentiable, the actions can receive gradients from multiple time steps ahead which can penalize actions that will lead to large diverges in the future even if the instantaneous divergence is small. Instead of learning a model, UrbanDriver (Scheel et al., 2022) constructs a differentiable data-driven model using recorded perception data and High Definition (HD) maps, where new observations are calculated by a coordinate transformation based on the pose of controlled SDV and collected data. However, the performance of the model-based approach is limited by the model's accuracy.

## 2.2 IMITATION LEARNING WITH ROBUSTNESS

IL is different from supervised learning by deploying the policy under dynamics, whose robustness is considered to be the learned policy's ability to recover from policy errors. Some researchers have attempted to learn a policy with a stability guarantee using control-theoretic methods by constraining policy and system dynamics. Taylor Series IL (Pfrommer et al., 2022) proves that the trajectory induced by the learner and expert will be close if their derivative difference at expert states is small but computing the high-order derivatives of the expert policy are difficult without sufficient data. Others learn a robust linear (Palan et al., 2020; Havens & Hu, 2021) or simple feed-forward neural network (Yin et al., 2021) control policy for a linear dynamical system by posing constraints on the policy. Even though the authors in East (2022) extend the robustness guarantee to the polynomial system and policy, obtaining guarantees on close-loop stability for the nonlinear autonomous driving system remains a challenge. Recently, the CMILe method (Tu et al., 2022) is capable of training nonlinear policies with the same safety guarantees as the expert but requires online expert access as DAgger. However, our method only constrains our policy in the formulation of receiving context state and producing ego state, whose stability is guaranteed under a mild input-to-state stability assumption on the environment dynamics.

## 3 THEORETICAL ANALYSIS

We first introduce the notations and definitions used in this paper. For any vector $\boldsymbol{x} \in \mathbb{R}^n$, $\|\boldsymbol{x}\|_p$ stands for its $L^p$ norm and $\|\boldsymbol{x}\|$ for $L^2$ norm. For any square matrix $\boldsymbol{A} \in \mathbb{R}^{n \times n}$, $\|\boldsymbol{A}\|$ denotes its induced $L^2$ norm and $\rho(\boldsymbol{A})$ is its spectral radius (the maximum of the absolute values of its eigenvalues). For every induced matrix norm, we have $\rho(\boldsymbol{A}) \leqslant \|\boldsymbol{A}\|$.

**Definition 1 (Comparison Functions)** *A function $\gamma : \mathbb{R}_{\geqslant 0} \to \mathbb{R}_{\geqslant 0}$ is class $\mathcal{K}$ if it is continuous, strictly increasing and satisfies $\gamma(0) = 0$. A function $\beta(x, t) : \mathbb{R}_{\geqslant 0} \times \mathbb{R}_{\geqslant 0} \to \mathbb{R}_{\geqslant 0}$ is class $\mathcal{KL}$ if it is continuous, $\beta(\cdot, t)$ is class $\mathcal{K}$ for each $t$ and $\beta(x, \cdot)$ is decreasing for each $x$.*

**Definition 2 (input-to-state stability (ISS) (Sontag, 2008))** *A discrete-time system $\boldsymbol{x}_{t+1} = f(\boldsymbol{x}_t, \boldsymbol{u}_t), \boldsymbol{x}_0 = \boldsymbol{\xi}$ is input-to-state stable if there exists a class $\mathcal{KL}$ function $\beta$ and a class $\mathcal{K}$ function $\gamma$ such that, for each bounded input $\boldsymbol{u}$ and initial condition $\boldsymbol{\xi}$ and $t \in \mathbb{Z}_+$, it holds that:*

$$\|\boldsymbol{x}_t(\boldsymbol{\xi}, \boldsymbol{u})\| \leqslant \beta(\|\boldsymbol{\xi}\|, t) + \gamma(\|\boldsymbol{u}\|_\infty), \tag{1}$$

*where $\|\boldsymbol{u}\|_\infty = \sup_{t \in \mathbb{Z}_+}(\boldsymbol{u}_t)$ is the sup norm of the input. If a system satisfies the ISS property, the distance between any two trajectories must eventually be bounded by its input signal and independent of initial conditions.*

We consider performing imitation learning in a nonlinear discrete-time system:

$$\boldsymbol{x}_{t+1} = f(\boldsymbol{x}_t, \boldsymbol{u}_t), \tag{2}$$

where $\boldsymbol{x}_t \in \mathbb{R}^n$ is the system state at time $t \in \mathbb{N}$ and $\boldsymbol{u}_t \in \mathbb{R}^u$ is the control input. For an autonomous driving system, its state can be separated into two parts $\boldsymbol{x}_t = (\boldsymbol{s}_t, \boldsymbol{c}_t)$: an SDV state $\boldsymbol{s}_t \in \mathbb{R}^m$ controlled by the control input $\boldsymbol{u}$ and context state $\boldsymbol{c}_t \in \mathbb{R}^{n-m}$ which is influenced by the SDV state. By viewing the policy error as a disturbance to the system, we apply linearization to the nonlinear system at the expert's state $\boldsymbol{x}^*$ to obtain the following linear subsystem:

$$\boldsymbol{c}_{t+1} = \boldsymbol{A}\boldsymbol{c}_t + \boldsymbol{B}\boldsymbol{s}_t, \tag{3}$$

with solution:

$$\boldsymbol{c}_{t+1} = \boldsymbol{A}^{t+1}\boldsymbol{c}_0 + \sum_{j=0}^{t} \boldsymbol{A}^{t-j}\boldsymbol{B}\boldsymbol{s}_j. \tag{4}$$

Here with abuse of notation, we use the same symbol $\boldsymbol{c}_t, \boldsymbol{s}_t$ for the context deviation $\boldsymbol{c}_t - \boldsymbol{c}_t^*$ and SDV state deviation $\boldsymbol{s}_t - \boldsymbol{s}_t^*$.

Because the real-world traffic system without the SDV state deviation $\boldsymbol{c}_{t+1} = \boldsymbol{A}\boldsymbol{c}_t$ is stable, we have $\rho(\boldsymbol{A}) < 1$. For such a Schur matrix, there are constants $c > 0$ and $0 \leqslant \sigma < 1$ such that $\|\boldsymbol{A}^t\| \leqslant c\sigma^t$ (Jiang & Wang, 2001). Besides, the context in traffic systems generally includes static map elements that are not influenced by SDV vehicles and intelligent human drivers who can quickly recover from the other vehicle's perturbation. It is natural to assume the subsystem in equation 3 to be input-to-state stable (ISS) with $\beta(r, t) < \sigma^{t-1}r$ and $\gamma(r) \leqslant \epsilon r$, where $\epsilon \geqslant 0$. Combining equation 1 and equation 4, we have:

$$\beta(r, t) = c\sigma^t r < \sigma^{t-1}r, \quad \gamma(r) = \sum_{j=0}^{\infty} c\sigma^t \|\boldsymbol{B}\| r = \frac{c\|\boldsymbol{B}\|}{1-\sigma} r \leqslant \epsilon r, \tag{5}$$

which implies that $c < \frac{1}{\sigma}$, and $\|\boldsymbol{B}\| \leqslant \frac{\epsilon(1-\sigma)}{c}$.

Next, we study how the policy in our formulation can be stable under this ISS system. We learn a context state feedback control $\boldsymbol{s}_t = \boldsymbol{u}_t = f_{\boldsymbol{\theta}}(\boldsymbol{c}_t)$ which directly maps the context state to the SDV state. For simplicity, we consider a linear policy $\boldsymbol{u}_t = \boldsymbol{K}\boldsymbol{c}_t$. Then, the linear subsystem in equation 3 can be simplified as $\boldsymbol{c}_{t+1} = (\boldsymbol{A} + \boldsymbol{B}\boldsymbol{K})\boldsymbol{c}_t$ whose stable condition is $\rho(\boldsymbol{A} + \boldsymbol{B}\boldsymbol{K}) < 1$. Thus, we have if $\|\boldsymbol{K}\| < \frac{c(1-c\sigma)}{\epsilon(1-\sigma)}$, then

$$\rho(\boldsymbol{A} + \boldsymbol{B}\boldsymbol{K}) \leqslant \|\boldsymbol{A} + \boldsymbol{B}\boldsymbol{K}\| \leqslant \|\boldsymbol{A}\| + \|\boldsymbol{B}\|\|\boldsymbol{K}\| \leqslant c\sigma + \frac{\epsilon(1-\sigma)}{c}\|\boldsymbol{K}\| < 1. \tag{6}$$

This result shows that the stability of closed-loop autonomous driving can be guaranteed as long as the context system is stable enough and the norm of our policy is sufficiently small. In practice, we will add a $L^2$ norm regularization into the training loss of our neural network policy to internalize this stability prior.

Finally, we show how the naive BC policy formulation which takes both the SDV state and context state and generates the displacement from the current state can fail with even a fixed context $\boldsymbol{c}_t = 0$. We consider a linear policy: $\boldsymbol{u}_t = \boldsymbol{K}_{bc}\boldsymbol{s}_t$ and update the state $\boldsymbol{s}_{t+1} = \boldsymbol{s}_t + \boldsymbol{u}_t$. Then, the dynamics

is $s_{t+1} = (I + K_{bc})s_t$. Because $\|I\| = 1$, we cannot obtain the stable condition $\rho(I + K_{bc}) < 1$ by limiting the $\|K_{bc}\|$ as our policy formulation. Thus, the system's stability cannot be guaranteed.

This simple analysis supports our choice to overlook the SDV history and current state in the learning to avoid causal confounding and distribution shift. Its advantage will also be validated in our experiments showing that this can significantly decrease the off-road rate.

## 4 METHOD

As shown in Figure 1, we apply our context-conditioned imitation learning method to learn a policy for urban driving. The policy network consists of a spatial encoder and a temporal encoder. The policy network is trained to minimize the $L^1$ distance between its predictions and the ground truth trajectories. During the evaluation, we employ a LQR controller to generate smooth planning based on the predicted trajectory from the policy network.

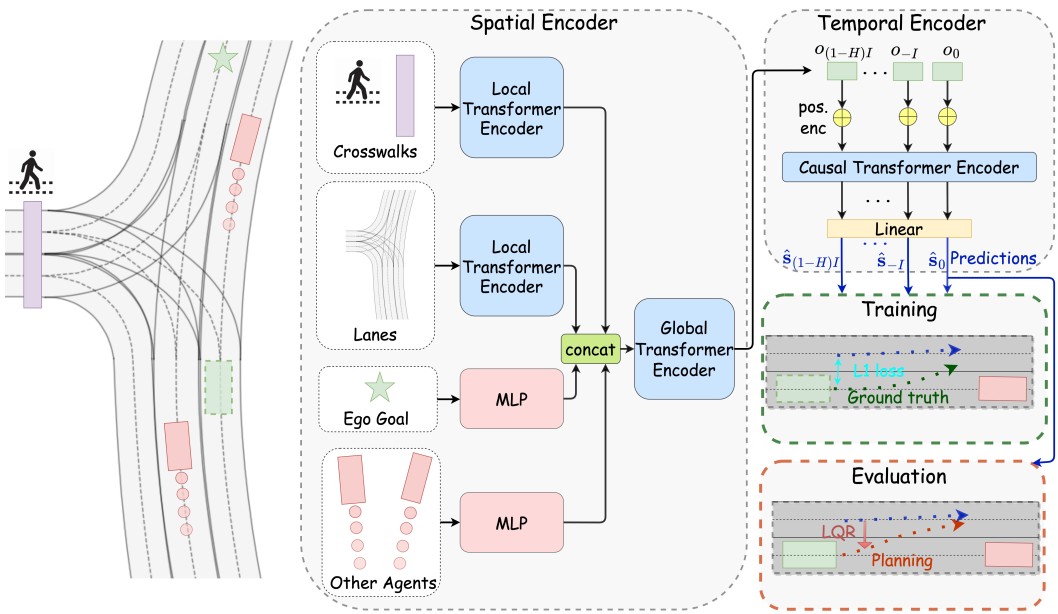

Figure 1: Overview of our approach

### 4.1 INPUT REPRESENTATION

Following Bansal et al. (2019), the intermediate representations such as oriented bounding boxes from a perception system and road networks from HD maps rather than raw sensor data (such as camera images or lidar points clouds) are harnessed to improve generalization and interpretability. In detail, there are two types of inputs to the policy network: map context and traffic participants. For the map context, we consider two types of map elements, which are crosswalks and lanes. Each map element is presented in a polyline which is a sequence of vectors with different associated features as Gao et al. (2020). The features of crosswalk vectors are their initial and terminal point's position and sequence orders, and lane vectors have additional features such as their traffic light state and lane width. For the traffic participants, we divide them into ego vehicle and other agents. For the ego vehicle, as discussed above, we should overlook the features which can be influenced by the learned policy to overcome the distribution shift obstacle. Therefore, we only input its mission goal position while removing its history. For each other agent, its features are composed of its type (e.g. car, bicycle, and cyclist), sizes, centroids, and orientations of bounding boxes at several past time steps.

Even though there is no explicit SDV state in the network input, the observations are typically translated to the SDV coordinate system using its rear axle's midpoint as origin and orientation

as x-axis direction in previous works (Bojarski et al., 2016; Scheel et al., 2022). This implicit incorporation of the SDV state can still cause a distribution shift of inputs. To reduce the implicit influence and ensure local observability, we adopt an ego-perturbed goal-oriented coordinate system whose origin is the SDV position adding a zero mean Gaussian perturbation, and $x$-axis direction is toward its goal position. Note that the perturbation is only required during training and the goal position can be replaced with any point that is not influenced by the driving policy.

## 4.2 POLICY NETWORK

The policy network is designed to directly plan a trajectory for the SDV using only its context. Since it is more challenging to plan without the historical trajectory, we need to make full use of the spatial-temporal interactions between the SDV and context to infer its future states. Thus, we first embed the observation at each time step into an observation feature by a spatial encoder, and then the observation features in current and previous time steps are embedded to generate the predictions by a temporal encoder. We construct our policy network based on the Transformer whose multi-head attention mechanism has demonstrated excellent performance in capturing relations.

**Spatial Encoder:** To capture the spatial relations between vectors of the same polyline such as a lane or crosswalk, we employ a local Transformer encoder as the vector-vector interaction encoder. Subsequently, we aggregate the features of vectors belonging to the same polyline by max-pooling to obtain polyline-level features. To obtain the agent and goal features, two multi-layer perceptions (MLP) are applied. To model the high-order interactions between the goal, agents, and map elements, we utilize a global Transformer encoder whose goal embedding is output as observation features.

**Temporal Encoder:** To take the temporal information and interaction into account, a Transformer encoder with a causal self-attention mask is employed to embed $H$ historical observation features with a step interval $I$. Then, each hidden state of the Transformer is decoded by a full-connected linear layer to generate a prediction of the SDV future state at future $T$ time steps. For simplicity, we represent the SDV state with its rear axle's midpoint coordinate and orientation.

## 4.3 TRAINING PROCESS

We leverage supervised learning like BC to train the policy network by minimizing the $L^1$ distance between the predictions and ground truth trajectories, as $L^1$ is more correlated to driving performance compared with the mean square error (Codevilla et al., 2018a). To help the network converge and generalize, we additionally introduce an auxiliary task of minimizing the $L^1$ error at all previous time steps and apply a squared $L^2$ norm regularization to the network parameters $\boldsymbol{\theta}$ inspired by equation 6. The final loss is:

$$L = \sum_{t=1}^{T} \left( \|\boldsymbol{s}_t - \hat{\boldsymbol{s}}_0^t\|_1 + \mu \sum_{h=1}^{H-1} \|\boldsymbol{s}_{t-hI} - \hat{\boldsymbol{s}}_{hI}^t\|_1 \right) + \frac{\lambda}{2} \|\boldsymbol{\theta}\|^2, \tag{7}$$

where $\boldsymbol{s}_t$ is the ground-truth SDV state at time step $t$ and $\hat{\boldsymbol{s}}_{hI}^t$ is the state prediction at the historical time step $hI$ for its future $t$ time step. $\mu$ is an auxiliary hyperparameter and $\lambda$ is a regularization factor.

## 4.4 EVALUATION PROCESS

Even though the prediction without SDV state input is more stable, it has difficulty ensuring the smoothness of the predicted trajectory from the current state. Prior works such as Vitelli et al. (2022); Amos et al. (2018) usually add a differentiable kinematic layer into the policy network to generate physically feasible planning. However, the differentiable kinematic layer will bring in the current SDV state which is undesirable for learning a policy. Therefore, we choose to obtain a smooth trajectory during evaluation by a linear-quadratic regulator (LQR) (Åström & Murray, 2021) which can efficiently minimize the total commutative quadratic cost of a linear dynamic system. We consider a finite-horizon, discrete-time linear system whose dynamics is described by:

$$\begin{bmatrix} \boldsymbol{p}_{t+1} \\ \dot{\boldsymbol{p}}_{t+1} \\ \ddot{\boldsymbol{p}}_{t+1} \end{bmatrix} = \begin{bmatrix} \boldsymbol{I} & \boldsymbol{D} & \boldsymbol{D}^2 \\ 0 & \boldsymbol{I} & \boldsymbol{D} \\ 0 & 0 & \boldsymbol{I} \end{bmatrix} \begin{bmatrix} \boldsymbol{p}_t \\ \dot{\boldsymbol{p}}_t \\ \ddot{\boldsymbol{p}}_t \end{bmatrix} + \begin{bmatrix} \boldsymbol{D}^2 \\ \boldsymbol{D} \\ \boldsymbol{I} \end{bmatrix} \boldsymbol{u}_t, \tag{8}$$

where $\boldsymbol{D}$ is a diagonal matrix with the time interval of each step as diagonal entries, $\dot{\boldsymbol{p}} = (\boldsymbol{v}_t, w_t)$, $\ddot{\boldsymbol{p}}_t = (\boldsymbol{a}_t, \alpha_t)$, $\boldsymbol{u}_t = (\boldsymbol{u}_t^p, u_t^\alpha)$ denote the positional and angular velocity, acceleration, and control respectively, subject to a quadratic cost function:

$$\mathcal{J} = \sum_{t=1}^{T} \left( \|\boldsymbol{p}_t - \hat{\boldsymbol{s}}_{0,t}\|_2^2 + w_w w_t^2 + w_a \|\boldsymbol{a}_t\|_2^2 + w_\alpha \alpha_t^2 + w_u \|\boldsymbol{u}_t^p\|_2^2 \right), \tag{9}$$

where $\boldsymbol{p}_t$ is regarded as the planning trajectory and the predicted states $\hat{\boldsymbol{s}}_{0,t}$ at the current step from the policy network are regarded as target poses; $w_w$, $w_a$, $w_\alpha$ and $w_u$ are the weights to balance the positional accuracy and the smoothness of the planned trajectory. At the end of the optimization, the LQR will output a smooth trajectory for the SDV to follow.

## 5 EXPERIMENTS

### 5.1 DATASET

To benchmark our method's performance, we conduct experiments on two real-world large-scale datasets: **Lyft Level 5 Prediction Dataset** (Houston et al., 2020): contains about 1000h urban driving demonstrations in Palo Alto, which have been separated into independent scenes of nearly 25s at 10Hz. We train our network on the provided 100h (16265 scenes) subset as Scheel et al. (2022) and test with all 16220 validation scenes. **nuPlan Dataset** (Caesar et al., 2021) provides an urban driving dataset with 1312h of human driving data from 4 cities (Boston, Pittsburgh, Las Vegas, and Singapore). Due to the huge difference in traffic rules and patterns in different cities, we extract driving data in Las Vegas as Phan-Minh et al. (2022). Then, we separate the data into independent scenes of 25s at 10Hz like Lyft and filter out the scene without a mission goal. After filtering, we obtain 63181 training, 4774 validation, and 6386 testing scenes. More details about the data prepossessing and corresponding model details are presented in the appendix.

### 5.2 CLOSED-LOOP EVALUATION

To evaluate the closed-loop performance of our method, we use a log-replay simulator as prior work (Scheel et al., 2022; Huang et al., 2022). At each step in the log-replay simulator, the SDV vehicle updates its state according to the planned trajectory, while the other agents are assumed to follow their recorded trajectories in the dataset. For both datasets, we evaluate our method for 25s at 10Hz with the following metrics:

**Collision Rate**: If the SDV collides with the other agents at any time step in a scene, the scene is deemed as a collision scene. The collision rate is the ratio of the collision scenes in all scenes.

**Off-road Rate**: In nuPlan, we use the official drivable area compliance metric, *i.e.* the distance of a corner of SDV's bounding box from the drivable area is more than 0.3m. But in Lyft, without access to the drivable area, if the SDV deviates laterally from the human driver's ground truth more than 2m like in a scene, the scene will be deemed off-road as Scheel et al. (2022).

**Discomfort**: we count the rate of the absolute value of the acceleration more than $3 \text{ m/s}^2$ over time steps to quantify the comfort and feasibility of the planned trajectory.

**L2**: we use the average L2 position errors between the roll-out trajectory and the human driver's ground truth to quantify the human driving similarity.

### 5.3 PERFORMANCE EVALUATION

On the Lyft dataset, we compare our methods against three state-of-the-art methods to demonstrate the advantage of the proposed framework:

**Raster-perturb**: an official BC planning baseline (Houston et al., 2020), which based on ResNet50 receives a Bird-Eye-View (BEV) representation of the scene surrounding the SDV and produces a trajectory of position and yaw displacements. To augment data, a perturbation is applied to the current SDV position, and then a new kinematically feasible trajectory to reach the original endpoint is generated as ChauffeurNet.

Table 1: Comparison with baselines of 25s closed-loop performance on Lyft and nuPlan dataset. There are no variances in Raster-perturb, BC-perturb, and UrbanDriver because we evaluate the deterministic pre-trained models in a deterministic simulator.

| Model | Num params | Collision(%) | Off-road(%) | Discomfort(%) | L2(m) |
|---|---|---|---|---|---|
| Raster-perturb | 23.6M | 15.48 | 5.06 | **4.00** | 5.90 |
| BC-perturb | 1.8M | 9.38 | 6.77 | 39.10 | 4.77 |
| UrbanDriver | 1.8M | 13.28 | 7.27 | 39.41 | 5.74 |
| TD3+BC | 2.8M | 22.53±1.76 | 15.21±0.97 | 4.86±0.47 | 6.34±0.41 |
| Vector-Chauffeur | 1.5M | 10.12±0.23 | 3.40±0.32 | 5.42±0.44 | 5.03±0.43 |
| **CCIL (ours)** | **1.5M** | **3.32**±0.15 | **0.62**±0.13 | 4.33±0.22 | **1.23** ± 0.08 |
| Raster | 23.6M | 62.45±3.45 | 32.15±1.95 | 21.50±3.19 | 22.71±2.20 |
| LaneGCN-perturb | 2.0M | 60.63±2.34 | 34.25±1.65 | 17.26±1.80 | 21.21±1.81 |
| TD3+BC | 2.8M | 39.12±2.21 | 18.59±1.04 | 10.56±0.95 | 15.04±1.62 |
| Vector-Chauffeur | 1.5M | 24.12±1.37 | 10.11±0.62 | 12.53±1.17 | 6.12±0.87 |
| **CCIL (ours)** | **1.5M** | **6.91**±0.11 | **3.08**±0.11 | **1.16**±0.05 | **3.68**±0.04 |

**BC-perturb**: a BC model provided by Scheel et al. (2022) with the same trajectory perturbation and output as **Raster-perturb** but its inputs are presented in vector formulation, which are processed by a VectorNet (Gao et al., 2020) and Transformer.

**UrbanDriver**: an offline policy gradient method proposed by Scheel et al. (2022) to imitate the expert's policy exploiting a differentiable data-driven simulator with the same data and model structure as **BC-perturb**.

On the nuPlan dataset, we compare our methods against two **official** baselines learned by BC due to a lack of other prior works:

**Raster**: a raster-based model that uses the ResNet-50 backbone to encode SDV, agent, and map information as raster layers to plan the SDV's trajectory.

**LaneGCN-perturb**: a vector-based model that uses a series of MLPs to encode SDV and agent signals and one LaneGCN (Liang et al., 2020) to encode vector-map elements and a fusion network to capture lane and agent intra/inter-interactions through attention layers. To augment data, the SDV trajectories are perturbed and other agents are randomly dropped out.

In addition to these models provided by prior works, we also learn our model by a representative method in behavior cloning and offline reinforcement learning on both datasets:

**TD3+BC** (Fujimoto & Gu, 2021): it adds behavior cloning term to the policy updating of Twin Delayed Deep Deterministic Policy Gradient (TD3) (Fujimoto et al., 2018) for implicit policy constraint. We construct the offline RL dataset by applying the trajectory perturbation augmentation and consider the collision and comfort for the reward design.

**Vector-Chauffeur**: our model learned using the same data augmentation method as ChauffeurNet including trajectory perturbation and ego past motion dropout. And we represent the data in the same coordinate system which uses the ego location as the origin and its heading perturbed by a uniform noise as the orientation.

The performance is shown in Table 1. It demonstrates that our method can outperform previous work significantly on both datasets. The collision rate on the nuPlan dataset is higher than the Lyft dataset because nuPlan's scenarios are more complex with more road agents.

## 5.4 ABLATION STUDY

The following ablation experiments on the Lyft dataset are used to expose the significance of different components of our model, whose results are shown in Table 2:

Table 2: Ablation experiments on 25s closed-loop performance on Lyft dataset

| Model | Perturb | Ego | Collision(%) | Off-road(%) | Discomfort(%) | L2(m) |
|---|---|---|---|---|---|---|
| w explicit ego | ✓ | ✓ | 20.29±0.88 | 19.18±2.98 | **0.57**±0.15 | 5.50±0.55 |
| w ego dropout | ✓ | ✓ | 14.05±1.53 | 5.02±0.88 | 0.63±0.20 | 4.15±0.47 |
| w ego coordinate | | ✓ | 11.31±1.44 | 9.79±1.34 | 0.95±0.05 | 3.87±0.11 |
| std=0 | | | 7.08±0.35 | 2.86±0.25 | 0.89±0.10 | 3.46±0.34 |
| std=1 | ✓ | | 3.39±0.17 | 1.00±0.16 | 1.99±0.15 | 2.10±0.05 |
| std=2 (ours) | ✓ | | **3.32**±0.15 | 0.62±0.13 | 4.33±0.22 | 1.23±0.08 |
| std=3 | ✓ | | 3.42±0.12 | **0.49**±0.10 | 7.35±0.26 | **0.91**±0.04 |
| w/o causal Trans | ✓ | | 4.28±0.25 | 1.43±0.25 | 6.53±0.26 | 1.63±0.10 |
| w/o LQR | ✓ | | 3.81±0.14 | 2.07±0.12 | 89.05±0.35 | 1.02±0.02 |
| w/o regularization | ✓ | | 4.07±0.16 | 1.05±0.14 | 4.96±0.30 | 1.23±0.09 |
| w/o auxiliary | ✓ | | 4.56±0.29 | 1.02±0.06 | 3.23±0.31 | 1.92±0.07 |

**Network input**: We first study the importance of removing explicit ego information from the network input. We consider two ways to bring the explicit ego input back. The first one is to directly input ego past positions and then process it similarly to the goal position. The other one is to additionally introduce a dropout of 50% at the ego input during training as Bansal et al. (2019). We can observe that in both explicit ego information input ways, there is a steep drop in the collision and off-road rate and L2 distance due to the covariate shift issue.

**Coordinate system**: To analyze the impact of the implicit ego information in the coordinate system, we first consider replacing our ego-perturbed goal-oriented coordinate system with the ego-centric coordinate system in ChauffeurNet using orientation uniformly around the heading. We observe that the ChauffeurNet coordinate system leads to inferior closed-loop performance. In our coordinate system, we add a Gaussian noise with zero mean to the ego current position to obtain the origin. Thus, we can increase the Gaussian noise's standard deviation to diminish the implicit ego information. We can observe that with the increasing of the standard deviation, the discomfort increases but the off-road rate and closed-loop L2 decreases, which implies that the implicit ego information may improve the instantaneous prediction accuracy but deteriorate the closed-loop performance.

**Architecture**: To demonstrate the importance of the temporal information and the effectiveness of the causal Transformer in capturing the temporal interactions, we replace it with a MLP. We find that the causal Transformer can achieve better performance. Besides, to show the importance of the LQR module in enhancing comfort, we ablate it. Removing the LQR mechanism leads to a huge drop in comfort.

**Loss**: To demonstrate the effectiveness of the additional term, we remove the auxiliary and regularization loss term separately. The result in Table 2 shows that they are both beneficial to improving performance.

## 6 CONCLUSION

We have proposed a new offline imitation learning method to mitigate the distribution shift of behavior cloning, where we learn a neural network to predict the SDV's future positions and orientations without SDV information by making full use of its constraints from and interaction with the context. Firstly, we analyze theoretically the stability of our policy formulation. In addition to removing the explicit SDV information input to the network, we present a new SDV-perturbed goal-oriented coordinate system for representing represent the observation input to remove the implicit SDV information and ensure local observability. We design a Transformer-based network to make full use of the history information to handle the more challenging learning task. Finally, we demonstrate the effectiveness of our approach in two real-world large-scale datasets with state-of-the-art performance.

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

## A MAP PREPOSSESSING

In Lyft, there are only two types of map elements: lane and crosswalk. In nuPlan, the map elements consist of a lane, lane connector, intersection, stop line, crosswalk, walkway, and car park. We categorize them into polyline elements including lanes and lane connectors, and polygon elements including stop lines, crosswalks, intersections, walkways, and car parks.

For each polyline element, it is approximated by a sequence of vectors with an interval of 3 meters. For both datasets, we prepossess the polyline map into a graph to take advantage of the topological information. We connect each vector with its nearest left, right, and next vector if exists. The nearest left or right vector means the nearest vector of its reachable left or right polyline. And the distance between vectors is represented by the Euclidean distance between their starting points. After building the vector graph, we can compute the travel distance between any two vectors using Dijkstra's algorithm. For both datasets, the polyline vector has shared features including coordinates of starting and ending point, distance to left and right vectors, distance to the mission goal, traffic light states (red or green), and the sequence order. However, lane vectors in Lyft have a lane width feature because the vectors approximate computed middle lines of lanes, while the lane or lane connector vectors in nuPlan have a lane left and width feature because the vectors approximate an annotated baseline. In addition, the polyline vectors in nuPlan have an additional speed limit and type features.

For each polygon element, we approximate it also with a sequence of vectors. In the Lyft dataset, the crosswalk vectors are directly constructed by connecting the original sequential annotation coordinates. But we approximate each polygon with fixed 20 vectors because the annotation point number is too large for elements like intersection. For both datasets, each polygon vector has features including coordinates of starting and ending points, and its order sequence, while vectors in nuPlan have additional type features.

The inputs to our neural network are composed of two types of map elements: polyline and polygon, and two types of agent elements: other agents and ego goal. The missing inputs are padded with zeros and masked out when calculating the attention. The origin is the perturbed SDV position.

**Polyline**: 30 topologically nearest polylines with vectors whose starting points are within 35m from the origin. The topological distance between the origin and a polyline is the minimum of topological distances between the origin and its vectors.

**Polygons**: 20 polygons whose boundaries are within 35m from the origin. If there are more than 20 polygons, they are selected according to the importance of their types: stop line, crosswalk, intersection, walkway, and car park.

**Agents**: the nearest 30 agents whose oriented boxes' centroids are within 50m from the origin. The agent features in Lyft include its centroid coordinates, yaws, shapes, types, and relative times in the past 2 and current steps. The nuPlan agents additionally have velocity features as they are provided.

**Goal**: the $x, y$ coordinates of the SDV's mission goal of a scene. In Lyft, the mission goal is not provided, so we regard the ending point of the lane where the SDV locates at the last time step of the scene as the mission goal. In nuPlan, we directly take the provided mission goal at the last time step of the scene as the scene mission goal.

## B MODEL

For both datasets. the same model architecture is used, whose hyper-parameters are listed in Table 3.

## C TRAINING

Our model is trained using the Adam optimizer with a learning rate of 0.0005, 10000 steps linear warm-up, $\beta = (0.9, 0.999)$, and batch size 128. We stop training after 30 epochs and select the model with smallest validation collision rate for evaluation. We train all models except the pre-trained model in Lyft dataset independently for 3 times and then report the mean and std of their performances.

Table 3: Hyper-parameters for both datasets

| Hyper-parameter | Value |
|---|---|
| Future steps $T$ | 15 |
| All Transformers dropout | 0.1 |
| All Transformers head number | 8 |
| All Transformers hidden size | 128 |
| Local Transformer layer number | 3 |
| Global Transformer layer number | 6 |
| Causal Transformer layer number | 3 |
| Causal Transformer length $H$ | 15 |
| Causal Transformer interval $I$ | 2 |
| Auxiliary weight $\mu$ | 0.3 |
| Regularization weight $\lambda$ | 0.0001 |
| LQR angular velocity weight $w_w$ | 0.1 |
| LQR acceleration weight $w_a$ | 0.1 |
| LQR angular acceleration weight $w_\alpha$ | 0.1 |
| LQR jerk weight $w_u$ | 0.1 |

## D EVALUATION

For Lyft baselines, we directly evaluate the pretrained model provided by the Lyft dataset and UrbanDriver. The Raster-perturb are model trained on train.zarr for 2 epochs at `https://github.com/woven-planet/l5kit/blob/master/examples/planning/train.ipynb`. The BC-perturb and UrbanDriver are from Open Loop and Urban Driver in `https://github.com/woven-planet/l5kit/blob/master/examples/urban_driver/train.ipynb`.

We evaluate all Lyft models from the second time step to compute current velocity using position information as input to the LQR with zero assumed initial acceleration.

For nuPlan baselines, we train the official models by ourselves using provided hyper-parameters and evaluate from the first time step because the velocity and acceleration information is provided.

## E RUNTIME

We conduct runtime experiments using a single Nvidia GeForce GTX 1080 GPU and an Intel i7-8700@3.2GHz CPU on Lyft dataset. We measure the runtime of each method its mean and std over all time-steps in an evaluation. The runtime results shown in Table 4 consider all components in each model including data-prepossessing, model inference and control. We observe that our architecture can achieve higher data processing efficiency and medium model inference efficiency compared with other methods. Our method takes longer total execution time due to the extra LQR control module which does not exist in prior works because they only focus on optimizing positional accuracy but not comfort. However, our approach can still be executed in real-time on this hardware.

Table 4: Averaged runtime per frame of individual components for each method

| Model | Data process (ms) | Model inference (ms) | Control (ms) | Total (ms) |
|---|---|---|---|---|
| Raster-perturb | 6.03±0.61 | **4.62**±0.16 | - | **10.65**±0.64 |
| BC-perturb | 6.69±0.72 | 6.78±5.26 | - | 13.47±5.65 |
| UrbanDriver | 6.33±1.41 | 12.78±8.80 | - | 19.11±9.23 |
| CCIL | **4.92**±0.63 | 6.74±0.34 | 11.09±0.27 | 22.75±1.07 |

## F  TOY EXPERIMENT

We design a toy experiment to vividly show our method's ability to reduce compounding error. In the experiment, we use synthetic data from a very ideal and simplified scenario where a SDV moves under ring road network with fixed 1 m/s at 1Hz, as shown in Figure 2. During training, the radius of the ring is a variable with a range from 10m to 100m and the circular lane are represented as a series of fixed lane points with same interval of nearly 1m.

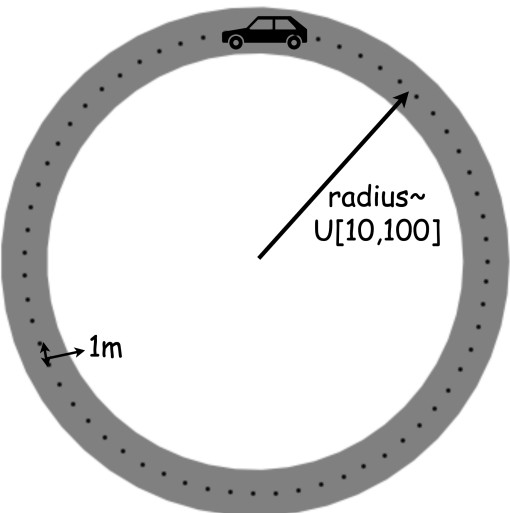

Figure 2: Ring road

We compare our CCIL methods with several baseline methods introduced above including BC, BC-perturb, UrbanDriver. The inputs of these baseline methods are composed of two parts: SDV state (its position and orientation in the past 10 time step) and context (nearest 10 lane points) in the ego coordinate system, while our methods only takes the context in the past 10 time step as inputs in the ego-perturbed center-oriented coordinate system. The ego-perturbed origin-oriented coordinate system means using the SDV position added a zero-mean one-std Gaussian perturbation as the origin and orientation to the center of the circular road as the x-axis direction. The trajectory perturbation is applied to augment the data in the BC-perturb method. For outputs, the BC and CCIL method generate the relative position and yaw at the next time step and the BC-perturb method produces the next 10 time steps. In the UrbanDriver method, we unroll the policy for 32 time steps.

In each methods, we employ a two-layer MLP with hidden size of 128 as a policy network. We train the neural networks using Adam optimizer of with a learning rate of 0.0001 with random initial weights for 100 times. We stop training after 10000 steps and then unroll the policy from one random starting point on the circle of radius 50m for 100 time steps. The 100 closed-loop trajectories for each method are depicted in Figure 3. We can observe that some trajectories in the baseline methods deviate from the road due to the covariate shift issue but the trajectories in our method keep following the route.

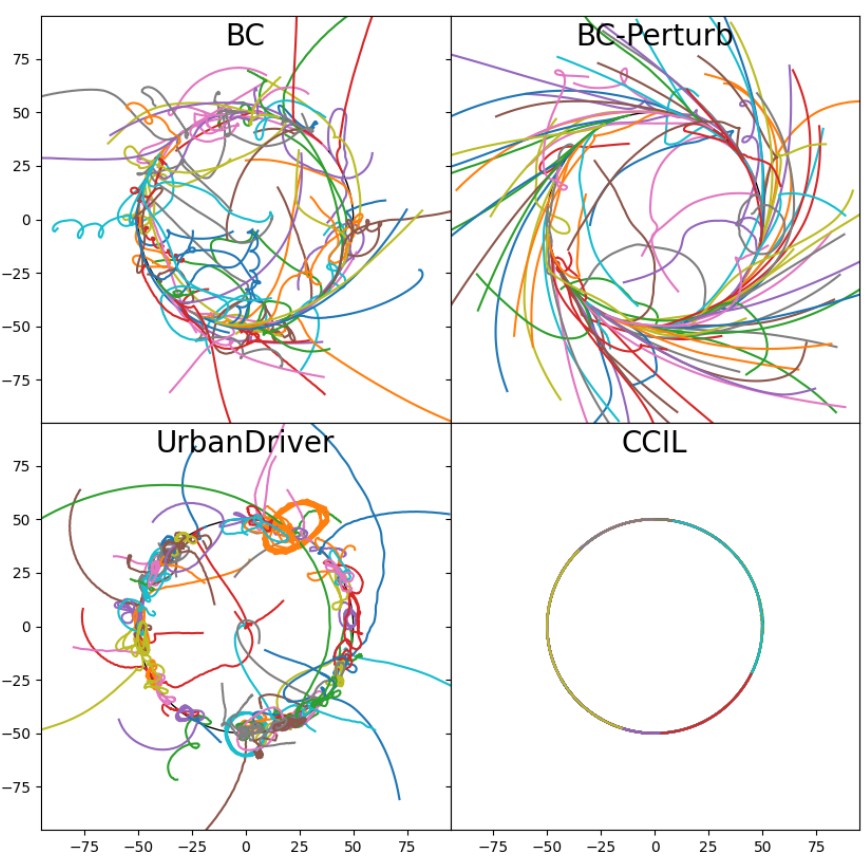

Figure 3: Closed-loop trajectories of each model trained on the toy dataset.

