# OpenReview forum: "CCIL: Context-conditioned imitation learning for urban driving"
_ICLR.cc/2023/Conference — Submitted to ICLR 2023_

### Official Review · Reviewer_oTYL · 2022-10-23

**Confidence:** 4
**Correctness:** 3
**Technical Novelty And Significance:** 2
**Empirical Novelty And Significance:** 2
**Recommendation:** 5

**Clarity, Quality, Novelty And Reproducibility:**

The novelty as limited as Past Motion Dropout was proposed in ChaufferNet.

The empirical evaluation is done in closed loop simulation only, and is not convincing. ChaufferNet showed that models trained with Past Motion Dropout are able to be deployed both in simulation and in real world. The proposed approach suggests that ChaufferNet policies could be trained without ego history, but only offers a limited evaluation in simulation.

The novelty of the Transformer architecture used for planning is also questionable, as it has been adopted in both motion forecasting and motion planning tasks.

Question to the authors: I did not understand what happens with the future ego trajectory after an initial state perturbation is applied. Is it updated as proposed in ChaufferNet?

**Strength And Weaknesses:**

Strengths:
- the paper is well written and is easy to follow

- it is great that the approach is evaluated on a publicly available planning benchmark

Weaknesses:
- there is no statistical significance evaluation of both the main comparison table and the ablation table. Planning benchmarks are known to have high variance, and it is not clear how training and evaluation randomness affect the results. I suggest the authors follow reinforcement learning experiments best practices, e.g. [1]

- baseline selection: it is explained how the model architectures differ and how many parameters they have. Would be great to add a column with numbers of parameters for every model in Table 1, for example:

|Benchmark|Model | Num params | Comment |
|:--|:--|:--:|:---:|
|Lyft|Raster-perturb | 25M | based on ResNet-50 |
|Lyft |BC-perturb | 2M | based on github colab |
|Lyft | UrbanDriver | 2M | or 3.5M in the paper, not clear |
|Lyft | CCIL | ? | ? |
|nuPlan|Raster | 25M | based on ResNet-50 |
| nuPlan |Vector-perturb | ? | |
| nuPlan |LaneGCN-perturb | ? | |
| nuPlan | CCIL | ? |  |

Based on the open-source code of UrbanDriver, it looks like BC-perturb and UrbanDriver models are small compared to ResNet-50 based Raster-perturb. Depending on the number of parameters in the proposed method, these approaches could be not directly comparable.

[1] Deep Reinforcement Learning at the Edge of the Statistical Precipice, NeurIPS 2021

**Summary Of The Paper:**

The authors propose a method based on the Past Motion Dropout hyperparameter from ChaufferNet, the behavioral cloning approach to autonomous vehicle motion planning. With the same motivation as ChaufferNet authors, they suggest to set the dropout parameter to 100%, such that neural network policy does not have access to past ego states. Similarly to ChaufferNet, they implement initial state perturbations as well. To compensate for the lack of temporal smoothness, the authors add a linear quadratic regulator. The authors also adopt a common Transformer-based architecture.

**Summary Of The Review:**

My suggestion is borderline due to minor novelty compared to ChaufferNet and the lack of real world testing. It is not clear how many parameters the proposed approach and selected baselines have, and if the results are statistically significant.

---

> ### Author Response · Authors · 2022-11-19
> **Reply**
>
> ## Q1: Similarly to ChaufferNet, they implement initial state perturbations as well.
>
> Thanks for your comment. Our perturbation is different from the perturbation in the ChauffeurNet in two aspects:
>
> - The perturbation in ChauffeurNet is applied to only initial time steps but we apply it to every time step in the inputs.
> - More importantly, the function of the perturbation in ChauffeurNet is to generate recovery examples for data augmentation. However, our perturbation is utilized to blur or conceal the implicit ego position information in the coordination system from the planning algorithm.
>
> ## Q2: there is no statistical significance evaluation of both the main comparison table and the ablation table. Planning benchmarks are known to have high variance, and it is not clear how training and evaluation randomness affect the results. I suggest the authors follow reinforcement learning experiments best practices, e.g. [1]
>
> Thanks for the suggestion. We have revised all results in the main comparison table and the ablation table into the mean±std formulation. We train each model three times except the pre-trained model provided by prior works. There is no evaluation randomness because the models and simulator are deterministic. As shown in these tables, the variances in our results are moderate.
>
> ## Q3: baseline selection: it is explained how the model architectures differ and how many parameters they have. Would be great to add a column with numbers of parameters for every model in Table 1.
>
> Thanks for your suggestion. We have added that column in Table 1. Our model can achieve the best performance with the smallest hyperparameter number.
> | Benchmark | Model |Num params |
> | ------ | ------ |------ |
> Lyft|	Raster-perturb|	23.6M	|
> Lyft|	BC-perturb	|1.8M	|
> Lyft|	UrbanDriver|	1.8M	|
> Lyft|	CCIL |	1.5M	|
> nuPlan|	Raster|	23.6M |
> nuPlan|	LaneGCN-perturb|	2.0M |
> nuPlan|	CCIL	|1.5M |
>
> ## Q4: The novelty as limited as Past Motion Dropout was proposed in ChaufferNet.
>
> Thanks for the comment. Our method is different from the Past Motion Dropout in three aspects:
> - during training, the dropout is sometimes (for example at 0.5 possibility) applied to the ego poses in ChaufferNet while we keep removing the ego poses completely;
> - during evaluation, the poses are always kept as input to the neural network in ChaufferNet while we remove them as in training;
> - in addition to removing the explicit ego poses, we design an ego-perturbed goal-oriented coordinate to remove the implicit ego information.
>
> We also consider an ablation method by applying the 0.5 motion dropout instead of removing them completely. The results are shown below:
> | Model | Collision(\%) | Off-road(\%) |  Discomfort(\%) | L2(m) |
> | ------ | ------ |------ |------ |------ |
> | w ego dropout | 14.05±1.53 | 5.02±0.88 | 0.63±0.20| 4.15±0.47  |
> | our | 3.32±0.15  | 0.62±0.13 | 4.33±0.22  |1.23±0.08  |
>
> Our method has a closed-loop performance significantly better than the dropout method.
>
> ## Q5: The empirical evaluation is done in closed-loop simulation only, and is not convincing. ChaufferNet showed that models trained with Past Motion Dropout are able to be deployed both in simulation and in the real world. The proposed approach suggests that ChaufferNet policies could be trained without ego history, but only offers a limited evaluation in simulation.
>
> Thanks for the comment. The real-world experiment is hard to deploy and benchmark. The closed-loop simulation is a common practice to validate planning methods in prior works [1,2].
>
> [1] Henaff, Mikael, Alfredo Canziani, and Yann LeCun. "Model-Predictive Policy Learning with Uncertainty Regularization for Driving in Dense Traffic." International Conference on Learning Representations. 2018.
>
> [2] Rezaee, Kasra, and Peyman Yadmellat. "How to not drive: Learning driving constraints from demonstration." 2022 IEEE Intelligent Vehicles Symposium (IV). IEEE, 2022.
>
> ## Q6: The novelty of the Transformer architecture used for planning is also questionable, as it has been adopted in both motion forecasting and motion planning tasks.
>
> Thanks for the comment. To our knowledge, our method is the first time to apply the Transformer both in spatial and temporal dimensions in motion planning tasks.
>
> ## Q7: I did not understand what happens with the future ego trajectory after an initial state perturbation is applied. Is it updated as proposed in ChaufferNet?
>
> We conduct two experiments on the Lyft dataset with our implementation of ChauffeurNet. One gives perturbed examples a weight of 0.1 relative to the real examples. The other one gives a weight of 1. We can observe a smaller weight leads to a small discomfort rate. It demonstrates that the perturbation leads to perturbed driving.
> | Model | Collision(\%) | Off-road(\%) |  Discomfort(\%)| L2(m) |
> | ------ | ------ |------ |------ |------ |
> | weight=0.1 | 10.12±0.23 |3.40±0.32| 5.42±0.44| 5.03±0.43  |
> | weight=1 | 12.13±0.25 |4.78±0.41| 8.96±0.65 | 5.75±0.51  |

---

### Official Review · Reviewer_VSrE · 2022-10-25

**Confidence:** 3
**Correctness:** 3
**Technical Novelty And Significance:** 3
**Empirical Novelty And Significance:** 2
**Recommendation:** 6

**Clarity, Quality, Novelty And Reproducibility:**

This paper is well written and well organized. The idea that removes all history ego vehicle's information from input to overcome covariate shift problem is full of novelty. I believe there will be more applications of this idea in the future.

**Strength And Weaknesses:**

Strength:
This paper proposes a novel offline learning method and a new coordinate system representation, which aims to mitigate the covariate shift problem. The proposed model outperforms the state of the arts in both Lyft and  nuPlan datasets in multiple matrics including collision, off-road, discomfort and L2 position errors with ground truth. In addition, the video and code are released, which will benefit the research of imitation learning. Besides, the paper is well written and has clear framework. In summary, the strength are novelty, performance and opensource.

Weaknesses:
1. Figure
Figure1:
Figure1 is not elegant, impressive or fresh. The authors really need to pay more attention to Figure1 and spend much more time on it.

Figure2:
I do not understand which method the authors choosed as "BC" method in figure2. The authors should explain more about Figure2 in details.
I do not understand why only one "BC" method was selected as comparison with proposed method in toy experiment either.
In Performance Evaluation the authors compared with three methods including Raster-perturb, BC-perturb and UrbanDriver. At least it is required to show the closed-loop trajectories of all these models.

2. Benchmark
Benchmark should take the state of the art offline reinforcement learning algorithms such as Conservative Q-Learning(CQL) into consideration, which makes the result more convincing.

**Summary Of The Paper:**

This paper proposed an offline method to overcome the covariate shift issue in imitation learning. A context-conditioned imitation learning method was proposed, which learns a policy to map context state into  ego vehicle's state without any history ego vehicle's information. To apply the method to urban driving, an ego-perturbed goal-oriented coordinate system was implemented, which helps to reduce the implicit ego information. Finally, model performance was tested on Lyft and nuPlan datasets with several baselines, which demonstrates proposed method can outperform all baselines.

**Summary Of The Review:**

Though there exist some problems in Figure and Benchmark, this paper is still considered as full of novelty and beneficial to the field. The paper is marginally above the acceptance threshold.

---

> ### Author Response · Authors · 2022-11-19
> **Reply**
>
> ## Q1: Figure1: Figure1 is not elegant, impressive or fresh. The authors really need to pay more attention to Figure1 and spend much more time on it.
>
> Thanks for the suggestion. We have polished figure 1 in the revised version.
>
> ## Q2: Figure2: I do not understand which method the authors choosed as "BC" method in figure2. The authors should explain more about Figure2 in details. I do not understand why only one "BC" method was selected as comparison with proposed method in toy experiment either. In Performance Evaluation the authors compared with three methods including Raster-perturb, BC-perturb and UrbanDriver. At least it is required to show the closed-loop trajectories of all these models.
>
> Thanks for the comments. We have added more details and comparisons with the BC-perturb and UrbanDriver methods on the toy dataset. Please refer to the last section of the appendix. The main difference between Raster-perturb and BC-perturb is the input representation, but this is fixed in this experiment. Thus, we ignore the Raster-perturb method.
>
> ## Q3: Benchmark should take the state of the art offline reinforcement learning algorithms such as Conservative Q-Learning(CQL) into consideration, which makes the result more convincing.
>
> Thanks for the suggestion. We have added a comparison with the TD3+BC method on both datasets because the TD3+BC works better on the datasets than CQL, which is hard to converge. The performance of offline RL is weaker than our method. The reason may be that most trajectories in the dataset of autonomous driving are nearly optimal and the collision reward is sparse.
>
> | Model | Collision(\%) | Off-road(\%) |  Discomfort(\%)| L2(m) |
> | ------ | ------ |------ |------ |------ |
> |TD3+BC(Lyft) | 22.53±1.76| 15.21±0.97 |4.86±0.47  | 6.34±0.41  |
> | CCIL(Lyft) | 3.32±0.15  | 0.62±0.13 | 4.33±0.22  |1.23±0.08  |
> |TD3+BC(nuPlan) | 39.12±2.21| 18.59±1.04| 10.56±0.95| 15.04±1.62|
> |CCIL(nuplan) | 6.91±0.11 |3.08±0.11| 1.16±0.05| 3.68±0.04 |\\

---

### Official Review · Reviewer_cGNi · 2022-10-25

**Confidence:** 3
**Correctness:** 3
**Technical Novelty And Significance:** 2
**Empirical Novelty And Significance:** 3
**Recommendation:** 5

**Clarity, Quality, Novelty And Reproducibility:**

Clarity: great, the paper is well written and the presentation is clear

Quality:good, attains SOTA numbers

Originality: not too great as mentioned in the above section


**Strength And Weaknesses:**

### Strengths
- The paper is well written and the presentation is clear.
- The proposed method is effective and achieves state-of-the-art performance. This is the strongest aspect of this paper in my opinion.

### Weaknesses
- I am not sure how helpful the theoretical analysis is. The assumptions in equation 3 and 6 which linearizes the state and policy does not match the real world or actual implementation of methods. I am also curious why the example at the end of section 3 particularly chooses the formulation of “u_t=s_t + K_bc s_t”.
- I am afraid that the proposed method has limited novelty. Using an ego oriented coordinate space for observation and predicting future states has been explored in behavior cloning for driving policy learning before [1,2] and is largely a standard practice in motion forecasting [3,4] (although the latter also contains ego state information as opposed to CCIL).
- I would like to see a latency analysis of the whole system as well as individual components of the methods, and how it compares with the baseline methods.
- I would like to see an ablation where the causal transformer encoder is trained to predict only the current step’s state predictions instead of all H steps.

[1] ChauffeurNet: Learning to Drive by Imitating the Best and Synthesizing the Worst, Bansal et al.

[2] TransFuser: Imitation with Transformer-Based Sensor Fusion for Autonomous Driving, Chitta et al.

[3] MultiPath: Multiple Probabilistic Anchor Trajectory Hypotheses for Behavior Prediction, Chai et al.

[4] Diverse Multi-Future Prediction and Planning for Self-Driving, Cui et al.


**Summary Of The Paper:**

This paper presents CCIL, a novel behavior cloning approach that learns a driving policy from human demonstrations. It adopts an ego-centric scene representation with a transformer-based architecture. The network predicts a sequence of future ego SDV states, which are converted to control actuations by a LQR. CCIL is evaluated in closed-loop planning driving benchmarks Lyft and nuPlan and attains state-of-the-art performances on both benchmarks.


**Summary Of The Review:**

The paper presents CCIL, a behavior cloning approach for driving policy learning. Although the presented approach is effective and attains state-of-the-art performance on both the Lyft and nuPlan benchmarks, aspects/main claims of the paper are not novel and heavily explored in driving policy learning/autonomous driving. Hence I do not recommend acceptance of the paper at the moment.

---

> ### Author Response · Authors · 2022-11-18
> **Reply**
>
> ## Q1: I am not sure how helpful the theoretical analysis is. The assumptions in equation 3 and 6 which linearizes the state and policy does not match the real world or actual implementation of methods. I am also curious why the example at the end of section 3 particularly chooses the formulation of “u_t=s_t + K_bc s_t”.
>
> Thanks for the comment. Applying linearization to a nonlinear system at an equilibrium point is a common practice in control-theoretic [1]. The correspondence between the analysis and the actual implementation is as follows. The expert state can be regarded as the equilibrium state. The policy error can be regarded as a small perturbation to the state. The Jacobian matrix of the nonlinear policy at the expert state is K.
>
> And we are sorry for the confusion caused by our typo. u_t=s_t+K_{bc} s_t should be u_t=K_{bc} s_t.
>
> [1] Khalil, Hassan K. Nonlinear control. Vol. 406. New York: Pearson, 2015.
>
> ## Q2: I am afraid that the proposed method has limited novelty. Using an ego oriented coordinate space for observation and predicting future states has been explored in behavior cloning for driving policy learning before [1,2] and is largely a standard practice in motion forecasting [3,4] (although the latter also contains ego state information as opposed to CCIL).
>
> Thanks for the comment. The novelty of our coordinate system is not the ego-centric part but the perturbation part. The coordinate systems in these works are all ego-centric. But using ego-centric coordinate can still implicitly leak ego information because the observation distributions in the ego-centric coordinate can be different given different choices of the coordinate origin, which can be abused by the planning algorithm to infer the ego information. This is a bit counter-intuitive, so an analogy with the non-inertial reference frame in physics can be helpful: in the ego-centric frame of a subject (like a car) that is moving with accelaration, it can figure out some movement information about itself from its local observation and thus the ego-motion is leaked. After noticing the harmful influence of such leaked implicit information on closed-loop performance, we design a coordinate system that is not influenced by ego motion, and this is our main contribution.
>
> Our proposed ego-perturbed coordinate system is just an expedient and exemplary way to reduce the leak of ego-motion information to the planning algorithm. Other coordinates not influenced by ego-motion can also be possible, such as using a unique lane point as origin, though it requires the vehicle to have more observation about the surrounding, which is not feasible in practice.
>
> ## Q3: I would like to see a latency analysis of the whole system as well as individual components of the methods, and how it compares with the baseline methods.
>
> Thanks for the suggestion. We conduct runtime experiments using a single Nvidia GeForce GTX 1080 GPU and an Intel i7-8700@3.2GHz CPU on the Lyft dataset. We measure the runtime of each method over all time steps in one evaluation. The result has been added to the paper:
> | Model | Data process(ms) | Model inference(ms) |  Control(ms)| Total(ms)|
> | ------ | ------ |------ |------ |------ |
> | Raster-perturb | 6.03±0.61  | 4.62± 0.16  | -   | 10.65±0.64 |
> | BC-perturb |  6.69±0.72 |  6.78±5.26 | -  | 13.47±5.65 |
> | UrbanDriver|  6.33±1.41| 12.78±8.80| - |19.11±9.23|
> | CCIL | 4.92±0.63 |6.74±0.34|11.09±0.27| 22.75±1.07 |
>
> We observe that our architecture can achieve higher data processing efficiency and medium model inference efficiency compared with other methods. Our method takes a longer total execution time due to the extra LQR control module. The control module does not exist in the other works because they only focus on optimizing positional accuracy but not comfort. However, our approach can still be executed in real-time on this hardware.
>
> ## Q4: I would like to see an ablation where the causal transformer encoder is trained to predict only the current step’s state predictions instead of all H steps.
>
> Thanks for the suggestion. We have added the ablation with the result shown below:
> | Model | Collision(\%) | Off-road(\%) |  Discomfort(\%)| L2(m) |
> | ------ | ------ |------ |------ |------ |
> | w/o auxiliary | 4.56±0.29 |1.02±0.06| 3.23±0.31| 1.92±0.07 |
> | w auxiliary | 3.32±0.15  | 0.62±0.13 | 4.33±0.22  |1.23±0.08  |
>
> Here, we regard the predictions of previous time steps as an auxiliary task. When the causal transformer encoder is trained to predict only the current step’s state predictions, the collision rate, off-road rate, and L2 will increase but the discomfort will decrease. The reason may be that the network will focus on the instantaneous prediction accuracy and then lose the generalization capability.

---

### Official Review · Reviewer_oLbQ · 2022-10-26

**Confidence:** 3
**Correctness:** 3
**Technical Novelty And Significance:** 3
**Empirical Novelty And Significance:** 3
**Recommendation:** 3

**Clarity, Quality, Novelty And Reproducibility:**

**Clarity, Quality:** the results look great but the contributions are not completely clear
**Novelty:** novel in terms of theoretical analysis of closed-loop planning
**Reproducibility:** code is provided and I believe it is reproducible

**Strength And Weaknesses:**

### Strengths

* Experiments show clear improvements over IDM, FLOW, COPO
* Overall mathematical notion is for the most part clear and well-motivated

### Weaknesses

* The main motivation seems quite counter-intuitive at first glance. Taking out ego-state seems like it would harm performance.
* Overall writing could be improved - several sections read off like lists.
* Figures could use some polish and better descriptions.
* nuPlan baselines are **extremely** weak which raises some concern.

### Minor Issues / Questions

* In table 2: it looks like w/o regularization is quite strong.
* How does the method perform without the LQR controller?

**Summary Of The Paper:**

This work builds a pipeline for closed-loop planning via behavior cloning.
Their propose to use only past local context (as opposed to SDV state and context in prior work), and use a transformer to predict future states, which is then turned to actions via LQR controller.
They show significant improvements over prior methods on the nuPlan and Lyft closed loop planning benchmark.

**Summary Of The Review:**

This work introduces a fairly counter-intuitive but effective approach and show significant improvements on closed-loop planning. However, I believe the paper does not have the appropriate amount of polish for the venue.

---

> ### Author Response · Authors · 2022-11-18
> **Reply**
>
> # Reviewer1
> ## Q1: The main motivation seems quite counter-intuitive at first glance. Taking out ego-state seems like it would harm performance.
>
> Thanks for the comments. Taking out the ego states during learning stages is reasonable, though indeed a bit counter-intuitive for the following reasons:
> - The ego states are the main source of the covariate shift in autonomous driving because a small error in learned policy can lead to significant deviation from the expert state distribution. It is highly likely that the historical trajectory induced by the learned policy has a gap with the distribution of the expert trajectory [1], because BC only matches the action distribution but not the trajectory distribution. Then, the tiny distribution shift in the inputs will lead to a large policy error. In practice, we observe a lot of off-road cases in the BC method which never occur in the dataset.
> - Taking out ego-states can also address the inertia problem stemming from causal confusion [2]. For example, when the ego vehicle stops, the probability it stays static is high in training data. This leads to a spurious correlation between low speed and no acceleration, inducing difficulty in restarting in the imitative policy. But our method masks the ego state, which makes it hard to build the spurious correlation.
> - Taking out the ego-state seems will only harm the instantaneous prediction performance (lower conformt) but will improve the long-term driving performance. We guess the reason can be that the ego-state can lead to temporal correlations in the prediction errors, because a consistent bias in one direction will lead to an off-road event while temporally uncorrelated noise may lead to slight oscillations around the expert trajectory but can result in successful driving. The phenomenon that offline prediction accuracy and actual driving quality are surprisingly weakly correlated is also found in [3].
>
> [1] Codevilla, Felipe, et al. "Exploring the limitations of behavior cloning for autonomous driving." Proceedings of the IEEE/CVF International Conference on Computer Vision (ICCV). 2019.
>
> [2] De Haan, Pim, Dinesh Jayaraman, and Sergey Levine. "Causal confusion in imitation learning." Advances in Neural Information Processing Systems 32 (2019).
>
> [3] Codevilla, Felipe, et al. "On offline evaluation of vision-based driving models." Proceedings of the European Conference on Computer Vision (ECCV). 2018.
>
> ## Q2: Overall writing could be improved - several sections read off like lists.
>
> Thanks for the suggestion. We have polished the writing to make the presentation clear.
>
> ## Q3: Figures could use some polish and better descriptions.
>
> Thanks for your suggestion. We have polished the figures and added more descriptions.
>
> ## Q4: nuPlan baselines are extremely weak which raises some concern.
>
> Sorry for the concern. The nuPlan baselines are weak because there is no prior work on this recently released dataset and thus we only utilize the baseline provided by the nuPlan official GitHub. To address your concern, we have compared our CCIL with two stronger baselines that have the same network structure as our method but are trained with the approaches in ChauffeurNet and TD3+BC, respectively.
> | Model | Collision(\%) | Off-road(\%) |  Discomfort(\%)| L2(m) |
> | ------ | ------ |------ |------ |------ |
> | TD3+BC | 39.12±2.21 |18.59±1.04| 10.56±0.95| 15.04±1.62|
> | Vector-Chauffeur | 24.12±1.67| 10.11±0.62| 12.53±1.17| 6.12±0.87|
> | CCIL | 6.91±0.11| 3.08±0.11| 1.16±0.05| 3.68±0.04 |
>
> ## Q5: In table 2: it looks like w/o regularization is quite strong.
>
> Thanks for the comment. That the w/o regularization is strong can result from the following reasons.
> - The norm regularization is important for linear policy based on the theoretical analysis but it is unclear whether the conclusion can be extended to a nonlinear policy represented in a neural network.
> - Our policy network has a modest parameter number so that the network is still well regularized even without the loss term.
> - The norm in our theoretical analysis needs to be an induced matrix norm. The L2 regularization corresponds to the Frobenius norm which is not an induced norm but only an upper bound of the spectral norm, i.e an induced matrix norm.
>
> ## Q6: How does the method perform without the LQR controller?
>
> Thanks for the comment. We add an experiment where the LQR controller is removed. The result is shown below:
> | Model | Collision(\%) | Off-road(\%) |  Discomfort(\%)| L2(m) |
> | ------ | ------ |------ |------ |------ |
> | w/o LQR | 3.81±0.14  | 2.07±0.12|  89.05±0.35  | 1.02±0.02|
> | w LQR | 3.32±0.15  | 0.62±0.13 | 4.33±0.22  |1.23±0.08  |
>
> We can observe that there is a small increase in the collision and off-road rate and huge increase in discomfort and a small drop in the L2. It demonstrates that the LQR is effective in enhancing the trajectory smoothness but the ego information taken in the LQR module can also harm the closed-loop L2 performance.

---

### Decision · Program_Chairs · 2023-01-20

**Decision:**

Reject

**Justification For Why Not Higher Score:**

The authors addressed many concerns of the reviewers. However, everyone (reviewers + AC) still thinks that the paper can benefit from one more iteration of experiments and validation.

**Justification For Why Not Lower Score:**

N/A

**Metareview: Summary, Strengths And Weaknesses:**

This work builds a pipeline for closed-loop planning via behavior cloning. The idea of setting the dropout parameter to 100% is not a new idea (suggested by ChaufferNet) and thus the methodological contribution is rather small. However, given the performance boost, I personally don't think "small" technical contribution is not an issue and also it is worth sharing with the community. The decision is still reject, mainly due to give one more polishing iteration in terms of visualization+writing, but more importantly analysis on where/why the performance gain is happening. This type of analysis can make up the lack of technical contribution.